

# Comparative analysis of bacterial communities associated with healthy and diseased corals in the Indonesian sea

Wuttichai Mhuantong[1,*], Handung Nuryadi[2,*], Agus Trianto[2], Agus Sabdono[2], Sithichoke Tangphatsornruang[3], Lily Eurwilaichitr[1], Pattanop Kanokratana[1] and Verawat Champreda[1]

[1] Biorefinery and Bioproduct Technology Research Group, National Center for Genetic Engineering and Biotechnology, Pathum Thani, Thailand
[2] Faculty of Fisheries and Marine Science, Diponegoro University, Semarang, Indonesia
[3] National Omics Center, National Center for Genetic Engineering and Biotechnology, Pathum Thani, Thailand
[*] These authors contributed equally to this work.

## ABSTRACT

Coral reef ecosystems are impacted by climate change and human activities, such as increasing coastal development, overfishing, sewage and other pollutant discharge, and consequent eutrophication, which triggers increasing incidents of diseases and deterioration of corals worldwide. In this study, bacterial communities associated with four species of corals: *Acropora aspera*, *Acropora formosa*, *Cyphastrea* sp., and *Isopora* sp. in the healthy and disease stages with different diseases were compared using tagged 16S rRNA sequencing. In total, 59 bacterial phyla, 190 orders, and 307 genera were assigned in coral metagenomes where *Proteobacteria* and *Firmicutes* were predominated followed by *Bacteroidetes* together with *Actinobacteria*, *Fusobacteria*, and *Lentisphaerae* as minor taxa. Principal Coordinates Analysis (PCoA) showed separated clustering of bacterial diversity in healthy and infected groups for individual coral species. *Fusibacter* was found as the major bacterial genus across all corals. The lower number of *Fusibacter* was found in *A. aspera* infected with white band disease and *Isopora* sp. with white plaque disease, but marked increases of *Vibrio* and *Acrobacter*, respectively, were observed. This was in contrast to *A. formosa* infected by a black band and *Cyphastrea* sp. infected by yellow blotch diseases which showed an increasing abundance of *Fusibacter* but a decrease in WH1-8 bacteria. Overall, infection was shown to result in disturbance in the complexity and structure of the associated bacterial microbiomes which can be relevant to the pathogenicity of the microbes associated with infected corals.

## INTRODUCTION

Coral reefs contain important ecological space harboring a diversity of marine organisms and represents the core element of the complex marine ecosystem. Southeast Asia contains the largest area of coral reefs, accounting for 34% of the world's total (*Wilkinson, 2008*).

Corresponding author
Pattanop Kanokratana, pattanopk@biotec.or.th

The reefs play crucial roles in ecology such as protecting the coastline from erosion, providing habitats for marine organisms, and supplying nutrients for the complex marine food chains. The coral reef ecosystem is estimated to provide USD 29.8 billion per year of economic benefit worldwide from various sectors, e.g., fisheries, tourisms, coastal protection, and biodiversity (*Cesar, Burke & Pet-Soede, 2003*). From a scientific point of view, corals are also considered a rich source of unique and unexplored biosynthetic products (*Radjasa et al., 2011*). In this ecosystem, diverse symbioses exist with complex interactive dependences between corals and associated communities of eukaryotic and prokaryotic microorganisms (*Blackall, Wilson & Van Oppen, 2015*). For example, dinoflagellate endosymbionts, Symbiodiniaceae (*LaJeunesse et al., 2018*)utilizes light energy (*Brodersen et al., 2014*) and secrete fixed carbon to the coral host (*Burriesci, Raab & Pringle, 2012*). This symbiosis creates a highly complex and unique ecosystem.

In the last few decades, coral reefs have been facing world-wide crisis related to coral degradation driven by complicated phenomena including anthropogenic stresses and natural factors (*Wilkinson, 2008*). It is estimated that the world has collectively lost 19% of the original area of coral reefs while 15% are seriously endangered with expected damage within the next 10–20 years and 20% are under threat of loss in 20–40 years (*Wilkinson, 2008*). Degradation of coral reefs ecosystem can lead to massive loss of marine biodiversity. One of the serious threats to coral reefs ecosystem comes from various coral diseases (*Harvell et al., 2007*; *Hughes et al., 2003*). The occurrence of these diseases has been increasing dramatically during the last decade due to the rising sea-water temperature as a consequence of climate change (*Rosenberg & Ben-Haim, 2002*). Many of these diseases are prevalent in reef-builder corals (Order Scleractinia), leading to the deterioration of entire reefs structures (*Harvell et al., 2007*; *Rosenberg et al., 2007*). Among them, black band, white band, white plague, white pox, brown band, red band, and yellow band syndromes are the main threats on corals in the Caribbean, Indo-Pacific, and Great Barrier Reef.

Coral diseases can be triggered by infection of pathogenic agents involving specific groups of bacteria, fungi, and viruses (*Bourne et al., 2009*; *Harvell et al., 2007*). Displacement of primary symbiotic microorganisms with other members has been shown to be related to the appearance of the disease signs. For example, the yellow blotch disease is caused by infection of *Vibrio* spp. which targets endosymbiotic zooxanthellae rather than the coral tissue, resulting in a decrease in chlorophyll concentration (*Cervino et al., 2004a*). Bacteria are considered as the major infectious microorganisms in corals. So far, only a limited number of bacteria have been isolated and identified as causative agents of coral diseases by Koch's postulates, including cyanobacteria for black band disease, *Vibrio* sp and *Acrobacter* sp for white band disease, *Vibrio* spp for yellow blotch disease and *Serratia marcescens* for white pox (*Sheridan et al., 2013*). However, the symbiotic nature of bacteria to corals makes them unculturable using current isolation techniques. Therefore, identification of the causative microbes by conventional culture-dependent approaches may not be able to give the complete scenario of coral associated microbial communities upon infection.

Culture-independent analysis of environmental communities through phylogenetic molecular markers allows direct study of the uncultured microorganisms, which can make up to 99% of the total diversity in many ecosystems (*Schloss & Handelsman, 2005*). Several
studies have shown that culture-independent approaches based on bacterial molecular markers have been performed to explore the microbial communities associated with healthy and corals under stresses in the Caribbean and Indo-Pacific regions and their roles in pathogenesis (*Gignoux-Wolfsohn & Vollmer, 2015*; *Meyer et al., 2017*; *Pootakham et al., 2018*). However, variation in the taxonomic composition of the microbial consortia related to existing coral diseases can depend on different geographical, seasonal, and physical factors of the coral habitats (*Sokolow, 2009*; *Woo et al., 2017*). In order to gain more information on the roles of uncultured bacteria on coral diseases in Southeast Asia, the bacterial communities associated with corals collected from Indonesian sea, which is considered one of the world's most important coral habitat, in healthy and diseased stages affected by black band disease (BBD), white band disease (WBD), white plaque disease (WPD),and yellow blotch disease (YBD) were explored in this study using tagged 16S rRNA sequencing on a next-generation sequencing platform. Our work compares coral-associated bacteria in different host species and shows shifts in the bacterial community structure during the diseased stage. Our findings expand the current understanding on the microbiology of these prevalent coral diseases.

## MATERIALS AND METHODS

### Sample collection and DNA extraction

Coral specimens were taken from the Indonesian sea in Central Java (Cemara Kecil (S05°49′57,7″E110°22′50,5″) and Karimunjawa (S05°50′12,9″E110° 25′14,1″))(Fig. S1) during September, 2015 from a depth of about 3–5 m, with the permission from the Balai Taman Nasional Karimunjawa (the approval number 934 /BTNKJ-1.6 /SIMAKSI /2015). Triplicate samples (10 g each) of fragments from healthy and lesion fragments from diseased corals were collected from individual coral species including *Acropora aspera*, *Acropora Formosa*, *Cyphastrea* sp. , and *Isopora* sp. Healthy and infected coral were collected from different colonies of same species on same reefs. Coral fragments were placed in sterile bottles filled with seawater obtained from the same sampling area during transportation. The samples were rinsed using sterilized sea water and preserved at −80 °C upon arrival at the laboratory (*Wegley et al., 2007*). Physicochemical properties of the seawater including temperature, salinity, dissolved oxygen and pH were measured using Lutron YK-2005WA (Lutron Electronic Enterprise, Taipei, Taiwan).

The coral specimens were grounded in a mortar prior to metagenomic DNA extraction using the SDS-based DNA extraction procedure (*Zhou, Bruns & Tiedje, 1996*), with slight modifications (*Kanokratana et al., 2004*). Briefly, 1 g of grounded sample was mixed vigorously for 30 min at 37 °C with 2.7 ml DNA extraction buffer (100 mM Tris-HCl (pH 8.0), 100 mM sodium EDTA (pH 8.0), 100 mM sodium phosphate (pH 8.0), 1.5 M NaCl, 1%(wt/vol) cetyltrimethylammonium bromide (CTAB)) and 20 µl of proteinase K (10 mg/ml). Subsequently,300 µl of 20% [wt/vol] SDS was added, and incubated at 65 °C for 2 h with gentle mixing every 15 min. After 2 h of incubation samples were centrifuged at $6,000\times$ g for 10 min at room temperature. The resulting supernatants were mixed with an equal volume of chloroform: iso-amyl alcohol (24: 1, vol/vol). The aqueous phase was transferred to a new

tube after centrifugation at 6,000× g for 10 min. To the aqueous phase, 0.6 volume of iso-propanol was added and the mixture was left at room temperature for 1 h, followed by centrifugation (14,000× g, 20 min). Then, the DNA pellet was washed with ice-cold 70% (vol/vol) ethanol and resuspended in double-distilled water. Healthy and diseased coral fragments were extracted separately. The integrity of the extracted DNA was verified by agarose gel electrophoresis. The concentration of extracted DNA was quantified using a NanoDrop ND-1000 spectrophotometer (Peqlab Biotechnologie GmbH, Erlangen, Germany).

## 16s rRNA amplification and sequencing

The purified metagenomic DNA was used as the template for amplification of the partial 16S rRNA gene using universal bacterial primers E785F (5′-GGATTAGATACCCTGGTAGTCC-3′) and E1081R (5′-CTCACGRCACGAGCTGACG-3′) encompassing the 5 and 6 hypervariable regions in prokaryotic 16S rRNA gene. Each pair of primers was attached with a specific 8-bp barcode sequence at 5′ end (*Meyer, Stenzel & Hofreiter, 2008*) and used for amplification of each metagenomic DNA sample in order to generate tagged 16S rRNA amplicons. Polymerase chain reactions were performed using Phusion DNA polymerase (Thermo Scientific, Espoo, Finland) on a MyCycler thermocycler for 25 cycles of denaturation at 98 °C for 10 s, annealing at 69 °C for 30 s, and extension at 72 °C for 30 s. The amplicons were purified using QIAGEN PCR purification kit (QIAGEN, Hilden, Germany) and were quantified using NanoDrop ND-1000 Spectrophotometer (Thermo Fisher Scientific, Waltham, MA, USA). Three independent PCR amplicons obtained from the individual coral metagenomic sample were performed. Sequencing was done using the ION PGM™ platform (Life Technologies, Carlsbad, CA, USA) following the manufacturer's recommended protocols.

## Data cleaning and analyses

Data analysis was done in QIIME version 1.9.1 (*Caporaso et al., 2010*) mainly for clustering sequences into operational taxonomic units (OTUs), classifying taxonomy and calculating diversity index. First, the raw sequences were initially demultiplexed into specific groups based on the tagged sequences and trimmed off low-quality score reads (Phred $\leq$ Q20), barcodes, and primers. Chimeric sequences and amplification errors were filtered by UCHIME (*Edgar et al., 2011*) using the referenced dataset from the Ribosomal Database Project (*Cole et al., 2014*). The OTUs were clustered from the remaining sequences with an "open-reference" method using UCLUST (*Edgar, 2010*) at 97% similarity. The representative sequence of each cluster was assigned for taxonomy against the Greengene database (*McDonald et al., 2012*). In order to compare the diversity among the samples, alpha diversity index including observed OTUs, Shannon-Weaver index, and the Chao1 richness estimator was calculated using a cutoff of 21,089 reads per sample, which was the minimal number of reads per sample obtained. The difference of microbial communities among samples was measured using weighted UniFrac distance (*Lozupone et al., 2011*) and Principal Coordinates Analysis (PCoA). The differences of bacterial adundance, OTU number, and diversity index between healthy and infected corals were statistically tested

**Table 1    Sampling locations and sea water characteristics.**

| Sampling location | Temperature (°C) | pH | Salinity[a] (PSU) | Dissolve oxygen (mg/l) | Turbidity (m) | Collected coral name |
|---|---|---|---|---|---|---|
| **Ujung gelam** (S05°49′57.7″, E110°22′50.5″) | 30.1 ± 0.4 | 7.3 ± 0.1 | 32.0 ± 0.4 | 6.4 ± 0.6 | 5 ± 0 | *Acropora formosa* |
| **Cemara Kecil** (S05°50′12.9″, E110°25′14.1″) | 28.3 ± 0.2 | 7.8 ± 0.1 | 34.3 ± 0.3 | 6.2 ± 0.3 | 2 ± 0 | *Acropora aspera Cyphastrea* sp. *Isopora* sp. |

Notes.
[a] Practical salinity unit.

using White's non-parametric $t$-test (*White, Nagarajan & Pop, 2009*) implemented in STAMP (*Parks et al., 2014*).

## RESULTS

### Sample characterization

Fragments of both healthy and diseased corals were collected from four major coral species in the sea of Central Java, Indonesia. Physicochemical characteristics of seawater in the sampling areas are described in Table 1. Both of the sampling sites have the same coral reef topology. Those sites are covered by various coral species from many lifeforms, such as branching, massive, submassive and encrusting. The seawater in Ujung Gelam had slightly higher temperature (30.1 °C) and dissolved oxygen (6.4 mg/l) but with lower pH (7.3) and salinity (32.0 PSU) than that in Cemara Kecil (28.3 °C, 6.2 mg/l, 7.8, and 34.3 PSU, respectively). The diseased corals were affected by different diseases commonly found in the Indo-Pacific sea. The physical appearances of the healthy and diseased corals are shown in Fig. 1 and Fig. S2. *A. aspera* infected by white band disease (*A. aspera*-WBD) showed symptom with the appearance of the white band between healthy and dead coral tissues. *A. formosa* infected by black band diseases (*A. formosa*-BBD) showed lesion as a reddish to dark-colored band. *Cyphastrea* sp. infected by yellow blotch disease (*Cyphastrea* sp.-YBD) which resulted in a band of yellow tissue around the enlarging sediment patch. *Isopora* sp. infected by white plaque disease (*Isopora* sp.-WPD) showed progressing band of bleached coral tissue followed by necrotic tissue starting from the base of the branch.

### Overview of sequencing dataset and alpha diversity analysis

The structures of bacterial communities associated with healthy and infected corals were investigated based on 16s rRNA sequences analyzed by the ION PGM platform. A total of 1,788,418 raw reads were obtained with an average read length of 260 base pairs. After the data cleaning process, 89.25% of raw reads (1,596,127 reads) were obtained ranging from 21,453 to 107,722 reads per sample. There was no significant difference of the average sequence number obtained from individual coral species in healthy and infected stages (ANOVA; $P > 0.05$). Total numbers of sequencing read for all samples are presented in Table 2 and Table S1.

Analysis of microbial community by alpha and beta diversity suggested the inequality of diversity among coral samples (Table 2). Overall, there was no significant difference in bacterial diversity between the two sampling sites or among the coral species. According

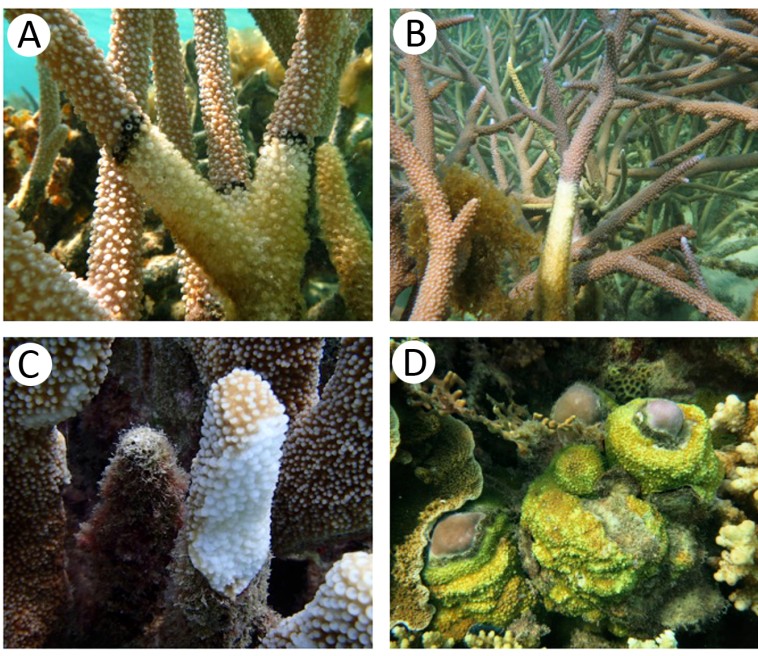

Figure 1 Appearance of coral specimens in healthy and infected stages; (A) black band disease, (B) white band disease, (C) white plaque disease, (D) yellow blotch disease.

to QIIME, the data set covered 96.8–99.1% of the bacterial diversity at 97% similarity. The number of OTUs at the 97% similarity ranged from 837 to 3,081 OTUs/ sample, which was lowest and highest in sample *A. aspera* (Aa-3) and *Isopora* sp. (Is-WPD-1), respectively. The highest total number of OTUs was found in sample *A. formosa* (Af-1) (4,440) while the lowest was observed in *Cyphastrea* sp. (Cs-YBD-1) (1,648) as estimated by the richness estimator, Chao1. In order to reduce the bias of unequal number of reads among the individual samples, normalization was performed by a subsampling method using a cutoff level of the minimal number of reads in an individual sample obtained (21,089 reads). Shannon's diversity indexes from OTU table (97% similarity) indicated significantly reduced diversity of bacteria in the diseased *A. formosa* and *Cyphastrea* sp. than healthy ones ($P$-value < 0.05); however, there was no significant difference in bacterial diversity between the healthy and diseased corals for species *A. aspera* and *Isopora* sp. ($P$-value > 0.05) (Fig. 2A). The shared OTU analysis revealed that the number of common OTUs in both healthy and diseased coral species ranging from 24.31–31.25%. A substantial fraction of OTUs in the healthy corals was replaced by newly introduced OTUs upon infection. This resulted in the presence of unique OTUs related to causative or opportunistic bacteria in all diseased corals, accounting for 28.78–49.51% of the total diversity of the coral-associated bacteria in the diseased stage (Fig. 2B).

The coverage of the dataset to the total diversity was assessed by rarefaction curve as shown in Fig. S3. For *A. formosa* and *Cyphastrea* sp., similar slopes were found for the healthy and infected corals, suggesting saturation of OTUs from both groups. In contrast, the slopes for the healthy *A. aspera* and *Isopora* sp. were substantially higher than that of

**Table 2** Diversity estimates of bacterial 16S rRNA gene in sequencing datasets of microbial communities associated with healthy and infected corals.

| Coral species | Sampling site | Condition | Sample ID | No. of sequences after QC | Observed OTUs | Chao1 | Shannon |
|---|---|---|---|---|---|---|---|
| *Acropora aspera* | Cemara Kecil | Healthy | Aa-1 | 102,886 | 1,732 | 3,096 | 5.5581 |
| | | | Aa-2 | 70,966 | 1,386 | 2,666 | 5.1340 |
| | | | Aa-3 | 24,556 | 837 | 1,913 | 5.0301 |
| | | Infected with white band disease | Aa-WBD-1 | 61,634 | 2,249 | 3,609 | 5.9713 |
| | | | Aa-WBD-2 | 107,722 | 1,948 | 3,337 | 5.4652 |
| | | | Aa-WBD-3 | 27,701 | 1,363 | 2,539 | 6.1005 |
| *Acropora formosa* | Karimunjawa | Healthy | Af-1 | 94,076 | 2,431 | 4,440 | 6.0624 |
| | | | Af-2 | 88,344 | 2,300 | 4,177 | 6.0045 |
| | | | Af-3 | 27,979 | 1,344 | 2,474 | 6.3581 |
| | | Infected with black band disease | Af-BBD-1 | 65,206 | 1,740 | 3,434 | 5.4460 |
| | | | Af-BBD-2 | 73,163 | 2,222 | 4,401 | 5.5205 |
| | | | Af-BBD-3 | 21,453 | 1,140 | 2,568 | 5.6828 |
| *Cyphastrea* sp. | Cemara Kecil | Healthy | Cs-1 | 55,264 | 1,793 | 3,479 | 6.1171 |
| | | | Cs-2 | 33,475 | 1,376 | 2,958 | 5.9524 |
| | | | Cs-3 | 77,176 | 2,212 | 3,764 | 6.2162 |
| | | Infected with yellow blotch disease | Cs-YBD-1 | 23,850 | 925 | 1,648 | 5.3805 |
| | | | Cs-YBD-2 | 63,703 | 1,987 | 3,769 | 5.5155 |
| | | | Cs-YBD-3 | 56,042 | 1,704 | 3,533 | 5.2922 |
| *Isopora* sp. | Cemara Kecil | Healthy | Is-1 | 55,471 | 1,592 | 3,324 | 5.6822 |
| | | | Is-2 | 103,771 | 2,033 | 3,611 | 5.6433 |
| | | | Is-3 | 102,458 | 2,252 | 3,555 | 6.0461 |
| | | Infected with white Plaque disease | Is-WPD-1 | 89,080 | 3,081 | 3,849 | 6.8630 |
| | | | Is-WPD-2 | 78,463 | 1,965 | 3,305 | 6.1575 |
| | | | Is-WPD-3 | 91,688 | 2,026 | 3,541 | 6.0657 |

the infected groups while the higher variation of OTUs was found in the infected corals compared to the healthy corals. The results thus suggested a high microbial adaptation of the bacteria associated with *A. aspera* and *Isopora* sp. in the infected stage.

## Taxonomic assignment of coral-associated microbial communities

Taxonomic classification by QIIME revealed that totally 59 bacterial phyla, 190 orders, and 307 genera were assigned in coral metagenomes. Only a small fraction (1.11%) of the total sequences could not be classified into any known phyla and was labeled as an unassigned sequence. According to the total dataset, the overall bacterial communities in the corals was dominated by sequences affiliated to the phylum *Firmicutes*, followed by *Proteobacteria* and *Bacteroidetes* as the major phyla across all samples, respective in their relative abundance, except in *A. aspera* where these three phyla showed relatively balanced distribution. Other phyla including *Actinobacteria*, *Fusobacteria,* and *Lentisphaerae* were present in lower abundances. Variation in their relative abundance was found depending on coral species and health status (Fig. 3). At the class level, *Clostridia* were a predominated class in *Firmicutes* while *Delta-* and *Gammaproteobacteria* were majorly presented in

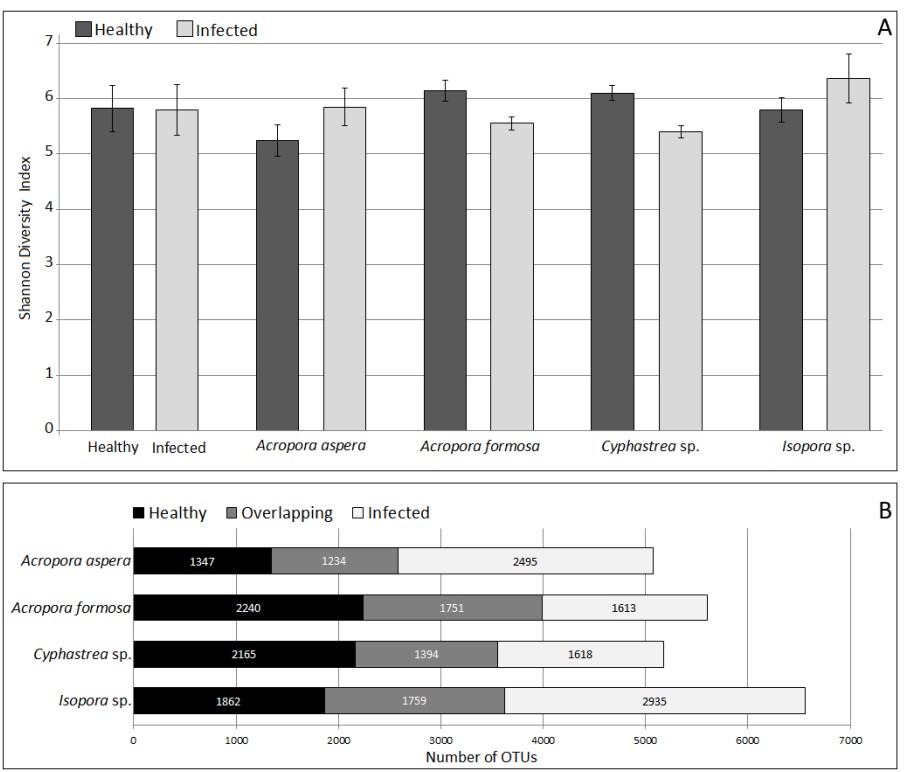

**Figure 2** **Comparison of Shannon's diversity indexes (A) and number of shared/unique OTUs (B) between healthy and infected conditions in each coral species.** Coral species marked with asterisk represented significant difference (White's non-parametric t-test) between healthy and infected corals.

phylum *Proteobacteria* in all coral samples, except in *Isopora* sp.-WPD. *Bacteroidia* and *Flavobacteria* existed as a major class in *Bacteroidetes* in all samples.

Comparative analysis based on coral's health status indicated that relative abundance of *Proteobacteria* was markedly increased in diseased corals *A. aspera*-WBD (50.18%) and *Isopora* sp.-WPD (39.14%) but slightly decreased in the other two coral species. With a higher number of *Delta-* and *Gammaproteobacteria* in *A. aspera*-WBD compared to the healthy one, a significantly decreasing number of *Flavobacteria* (from 14.48% to 2.44%) in infected *A. aspera*-WBD was clearly observed while an abundance of *Clostridia* was not significantly different ($P$-value > 0.05) in both healthy and infected groups. Higher relative abundances of *Delta-*, *Epsilon-* and *Gammaproteobacteria* in *Isopora* sp.-WPD were observed in agreement with the reduction of *Clostridia* when compared to the healthy corals. The ratio of *Clostridia* was significantly higher in *A. formosa* -BBD and *Cyphastrea* sp.-YBD compared to the healthy ones.

Focusing at the deeper taxonomic level, different relative abundances with significant variations in some bacterial genus were observed depending on coral species and health status (Fig. 4). *Fusibacter* belonging to phylum *Firmicutes* was predominantly present across all samples except *A. aspera*-WBD, where *Vibrio* was a major genus with approximately two times higher than the *Fusibacter*. This was related to the marked decreasing relative
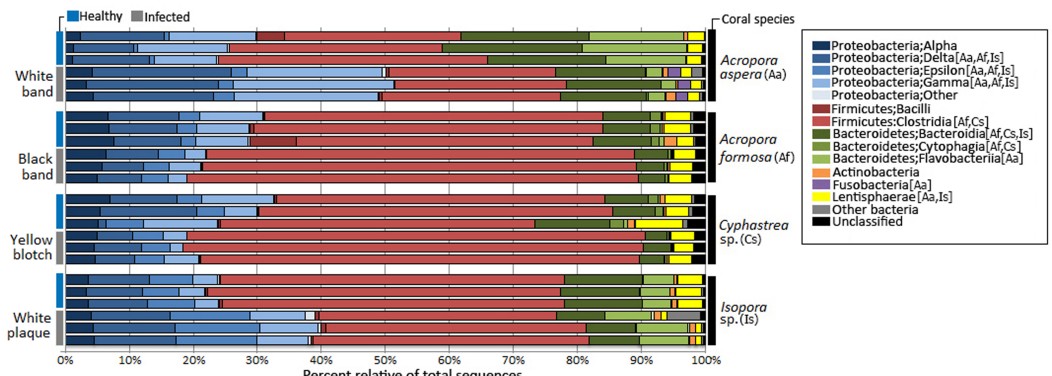

**Figure 3** **Relative abundance of bacterial distribution profiles at class level within the coral metagenomic samples obtained from healthy and diseased coral specimens.** Bacterial taxa labelled with coral species; Aa, Af, Cs, and Is indicated significantly difference (White's non-parametric t-test) between healthy and infected corals.

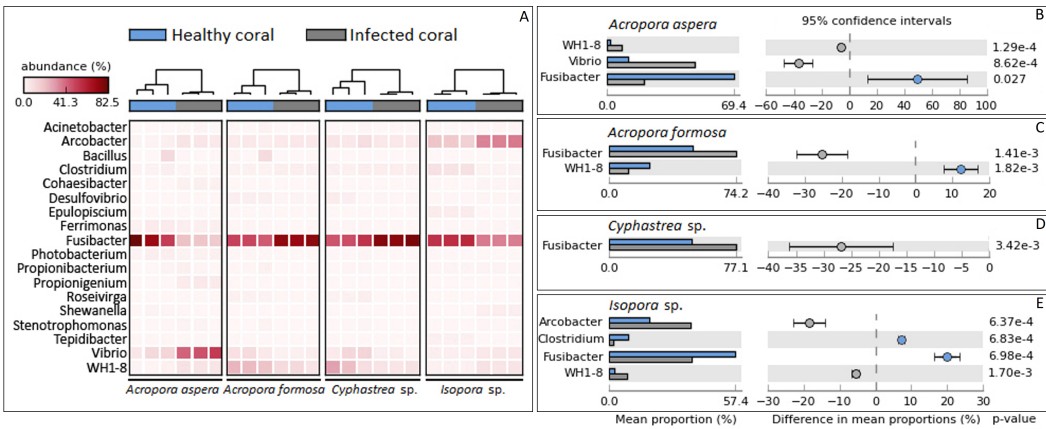

**Figure 4** **Taxonomic classification of bacterial diversity at the genus level.** Major bacterial genera were selected and shown as (A) a heat map based on the coral species and their health status and (B –E) difference in mean proportions healthy and infected corals.

abundance of *Fusibacter* in *A. aspera*-WBD along with a significantly increased ratio of *Vibrio* (and *Acrobacter* but not significant). A lower number of *Fusibacter* in infected corals was also found in the *Isopora* sp.-WPD, but the number of *Vibrio* was not changed significantly ($P$-value $> 0.05$) regardless of the coral health. Relative abundance of *Arcobacter* in class *Epsilonproteobacteria* was significantly higher ($P$-value $< 0.05$) in *Isopora* sp.-WPD but was not different in the other groups of diseased coral. On the other hand, the amount of *Fusibacter* found in the *A. formosa*-BBD and *Cyphastrea* sp.-YBD was higher than that found in the healthy ones ($P$-value $< 0.05$). This was associated with the increase of genus WH1-8 within phylum *Firmicutes* in the *A. aspera*-WBD and *Isopora* sp.-WPD samples.

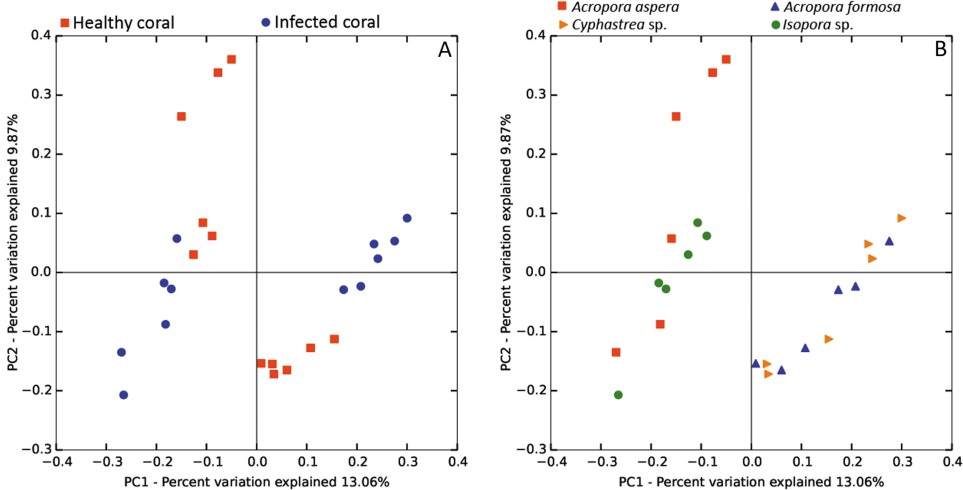

**Figure 5** Weighted UniFrac PCoA plot showing the differences between bacterial communities based on (A) the coral health status and (B) coral species.

## Correlation of bacterial diversity and beta diversity analysis

The beta diversity of bacterial communities related to the coral health status was measured using the weighted Unifrac method. Principal Coordinates Analysis (PCoA) based on weighted distance metric indicated that the bacterial diversity in healthy and diseased coral samples were clustered separately from each other for individual coral species (Fig. 5A). This suggested alteration of bacterial diversity when the corals were infected by any of these diseases. Distribution of bacteria was also clustered according to health status of corals regardless of the coral species, i.e., clusters between (1) healthy *A. formosa* and *Cyphastrea* sp., (2) infected *A. formosa*-BBD and *Cyphastrea* sp.-YBD, (3) healthy *A. aspera* and *Isopora* sp., and (4) infected *A. aspera*-WBD and *Isopora* sp.-WPD (Fig. 5B). Moreover, the result from PCoA plot conforms to the UPGMA tree constructed from weighted UniFrac distances (Fig. 6). The UPGMA tree topology showed clearly separated clusters between healthy and infected coral samples including (1) healthy *A. aspera* and infected *A. aspera*-WBD, (2) healthy *Isopora* sp. and infected *Isopora* sp.-WPD, and (3) healthy *A. formosa*, *Cyphastrea* sp. and infected *A. formosa*-BBD, *Cyphastrea* sp.-YBD. As individual coral species used in this study displayed different symptoms of diseases, the results suggested that when different coral species containing similar bacterial profiles were infected with different diseases, bacterial communities tend to alter to the similar direction. For example, bacterial profiles between *A. formosa* and *Cyphastrea* sp. were closed in the health status. However, when they were infected by BBD (*A. formosa*-BBD) and YBD (*Cyphastrea* sp.-YBD), bacterial communities in these two coral species were changed to the same direction and clustered together.

Changes in bacterial diversity in the healthy and infected corals are shown by a box plot of Phylogenetic Diversity (PD) Whole Tree. According to the results (Fig. S4), there is no significant difference in bacterial diversity measured by PD between healthy and infected corals ($P$-value > 0.05). Interestingly, the variation of PD values in diseased corals tended

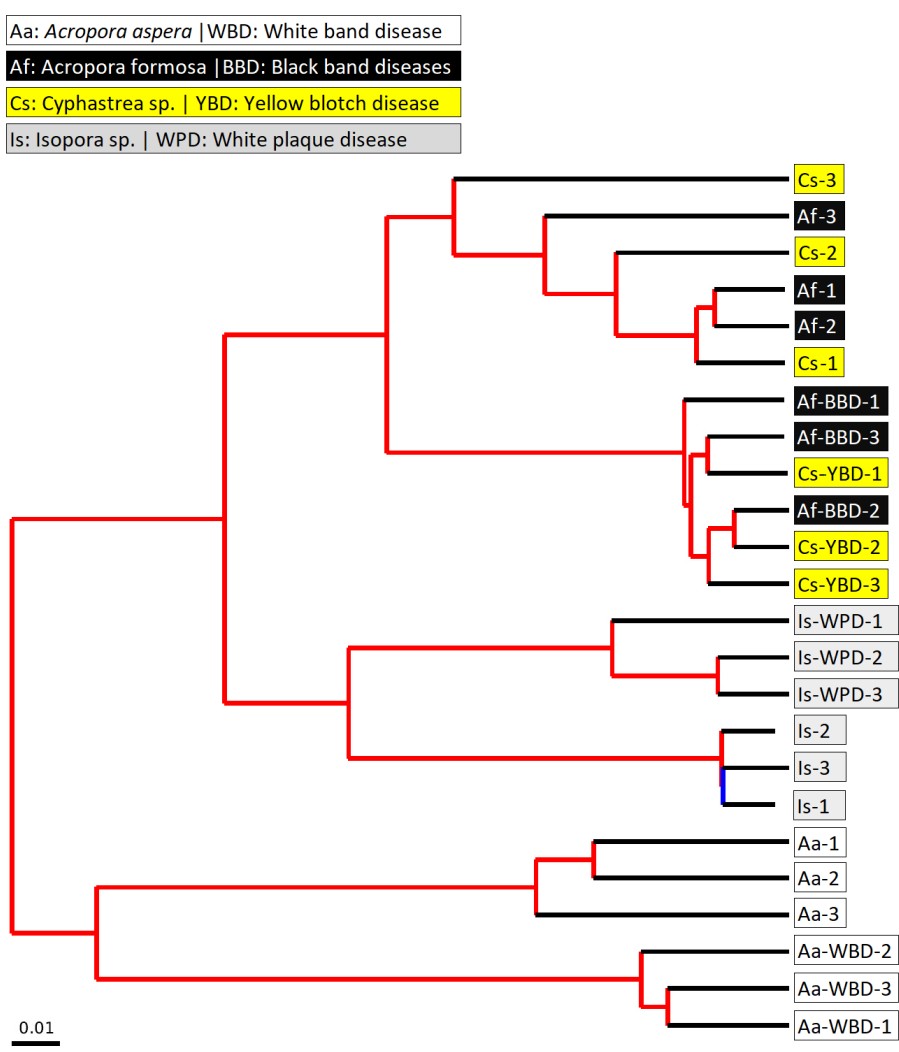

| Aa: *Acropora aspera* | WBD: White band disease |
| Af: Acropora formosa | BBD: Black band diseases |
| Cs: Cyphastrea sp. | YBD: Yellow blotch disease |
| Is: Isopora sp. | WPD: White plaque disease |

**Figure 6** **Jackknifed UPGMA tree of all coral samples based on weighted UniFrac distance matrix.** The jackknifed bootstrapped tree was illustrated with internal nodes colored; red for 75–100% support, yellow for 50–75%, green for 25–50%, and blue for < 25% support.

to be higher than the healthy coral possibly because the diseased coral is more susceptible to colonization by opportunistic bacteria. This was associated with higher or lower in diversity of the associated bacteria. Infection of *A. aspera* with WBD resulted in a marked increase in diversity of the bacterial communities. This situation was also observed in *Isopora* sp. and *Isopora* sp.-WPD. Decrease in complexity of the communities was found in *A. formosa*-BBD and *Cyphastrea* sp.-YBD. The results thus suggest alteration in the community structure of coral associated bacteria upon infection.

## DISCUSSION

Coral diseases are the result of complex interactions between host, causative agents, and environment (*Martin, Meek & Willeberg, 1987*; *Sunagawa, Woodley & Medina, 2010*).

They are characterized by a shift in microbial communities in coral mucus and tissue. However, causes and consequences of this phenomenon to pathogenesis is usually not fully understood due to the complexity and dynamics of the associated bacteria as well as effects of abiotic factors. According to our study, shifts in bacterial communities were found in all taxonomic levels upon infection of all coral species by all diseases. WBD is an important coral disease causing loss of corals in many regions of the world. It has been reported to exhibit high host specificity, particularly *Acropora* species, including *A. cervicornis* and *A. palmata* in the Caribbean (*Kline & Vollmer, 2011*). Based on the findings in our work, *A. aspera* showed a reduced diversity index for its associated bacteria upon infection by WBD. This was related to the increasing abundance of all sub-phyla of *Proteobacteria* along with the decreases in *Firmicutes* (*Clostridia*) and *Bacteroidetes* (*Flavobacteria*) with the presence of *Fusibacteria* found only in the diseased stage. This phenomenon occurred along with a significant increase in *Vibrio* and WH1-8 and decreasing *Fusibacter* upon infection. The causative agent of WBD is currently unknown but has been shown to possibly cause by bacteria according to a study on *A. cerevicornis* and *A. palmata* using antibiotic treatment (*Sweet, Croquer & Bythell, 2014*). *Vibrio* spp. were consistently found in association with both healthy and diseased corals (*Cunning et al., 2008*; *Mouchka, Hewson & Harvell, 2010*). Comparison of bacterial diversity in healthy and WBD-infected *A. palmata* based on 16s rRNA showed decreasing relative abundance of *Betaproteobacteria* and *Actinomycetes* with increasing abundance of *Planctomycetes* and *Cyanobacteria* (*Pantos & Bythell, 2006*). A putative pathogen *V. charcharii* has been identified and partially proven as the causative agent for WBD type II based on Henle-Koch's postulate (*Kline & Vollmer, 2011*). However, the causative bacteria for WBD type I has not yet been identified.

BBD is a polymicrobial coral disease which is considered one of the most virulent disease of scleractinian corals caused by polymicrobial factors which result in massive destruction of framework-building corals worldwide. Although factors affecting susceptibility of corals to BBD are still not fully understood, a few works showed that BBD pathogenesis is linked to nutrient enrichment, elevated temperature and light intensity. A recent study on Caribbean corals showed that the microbial communities in heathy corals were dominated by *Gammaproteobacteria*, particularly *Halomonas* spp. while the microbiome of BBD consortia were more variable and diverse (*Meyer et al., 2017*). Studies using a culture-independent molecular approach showed a diverse microbial community classified into four functional groups, including photoautotrophs (*Cyanobacteria*), sulfate-reducers (*Desulfovibrio*), sulfide oxidizers (*Beggiatoa*) and organo-heterotropths (*Vibrio*) (*Sere et al., 2016*). Among them, *Desulfovibrio* spp. and *Vibrio coralliilyticus* were suspected as the primary pathogens; however, without proven by Henle Koch's postulate (*Sere et al., 2016*). Basically, the complex microbial consortia act to produce highly concentrated sulfide specifically by the promoted Deltaproteobacteria under anoxic conditions beneath the BBD mat that are lethal to coral tissue (*Sato et al., 2017*). The accumulation of sulfide underneath the BBD mat was partially due to the lack of sulfur oxidizers which contributes to the lethality of the disease (*Meyer et al., 2017*). Meta-analysis of published clonal library studies of BBD microbial communities showed that, with few exceptions, the microbial species composition of BBD communities did not correlate with the species of the host corals

with the domination of OTU of *Roseofilum reptotaenium* in over 70% of the samples (*Miller & Richardson, 2011*). In addition, three OTUs of Bacteroidetes and Alpha-proteobacteria were present in 13% of the samples with other OTUs found in <7% of the samples.

According to our study, decreasing relative abundance of *Proteobacteria*, particularly the subphyla *Epsilon* and *Gammaproteobacteria,* was found with increasing *Firmicutes* (*Clostridia*) in the *A. formosa*-BBD. This was related to increasing abundance of *Fusibacter* and WH1-8 bacteria upon the infection. Increasing abundance of *Fimicutes, Cytophaga-Flexibacter-Bacteroidetes* (CFB) and *Deltaproteobacteria* in the infectious mat was identified in a pioneered molecular analysis work using Terminal Fragment Length Polymorphism (T-RFLP) (*Frias-Lopez et al., 2004*). The results showed that the composition of the infectious bacterial mat was not related to the species of coral being infected. Instead, differences in the mat composition appear to be linked to the species of cyanobacteria dominant in the infection. However, the presence of cyanobacteria in *A. formosa*-BBD could not be detected due to the specificity of primers used for amplification (*Nübel, Garcia-Pichel & Muyzer, 1997*). No increase in *Desufovibrio* was observed in the 16S rRNA sequencing analysis of *A. formosa*-BBD in this study. According to a previous study (*Meyer et al., 2016*), *Desulfovibrioprofundus* which is thought to be responsible the production of $H_2S$ was detected in only 5% of the clone libraries analyzed. These discrepancies could be due to technical issues (e.g., amplification biases and low coverage of microbiome in clonal libraries), seasonal and/or regional differences in the BBD composition or function-based (rather than taxonomic-based) of the BBD community. Variations have been detected in bacterial communities associated across geographical regions and between sympatric coral species (*Sere et al., 2016*). High variability in the BBD bacterial communities in different geographical areas and coral species suggested that this disease derives from an earlier infection, which aids subsequent infection of opportunistic microorganisms such as cyanobacteria (*Sere et al., 2016*). Recent analysis of metageomes from Caribbean and Pacific BBD mat revealed five metagenome-assembled genomes of *Roseofilum*, *Proteobacteria* and *Bacteroidetes* which are proposed to play symbiotic interaction (*Meyer et al., 2017*). However, the mechanisms of BBD development remain unclear, and no primary pathogens have yet been identified. The difference on the results observed in this study to previous works could be due to the difference of environmental factors such as climatic condition and location (*Mouchka, Hewson & Harvell, 2010*).

YBD infection in corals resulted in a decrease in microbial diversity index in *Cyphastrea* sp.. This was along with reduction in the relative abundance of *Proteobacteria* and *Bacteroidetes* (*Cytophaga*) and increase in *Firmicutes* (*Bacilli*) along the infection. *Vibrio alginolyticus* has been identified as a causative agent for YBD (*Cervino et al., 2004b*). Four Vibrio species (*V. rotiferianus*, *V. harveyi*, *V. alginolyticus*, and *V. proteolyticus*) were identified as causative agents in YBD in Caribbean corals through a series of infection and isolation experiments (*Cervino et al., 2004b*; *Cunning et al., 2008*). It is suggested that this Vibrio consortium infected the coral's symbiotic algae and resulted in degradation of zooxanthellae leading to the pale-yellow bands observed on infected Caribbean and Indo-Pacific corals (*Cervino et al., 2008*; *Cunning et al., 2008*). However, the pathogenesis mechanism during YBD infection is still poorly understood. A survey of *Vibrio* species
associated with healthy corals and YBD infected corals were conducted using a culture-based approach (*Cunning et al., 2008*). The results showed a shift from isolates taxonomically affiliated with *V. fortis* dominate in healthy corals to those related to *V. harveyi*, a known marine pathogen in diseased corals. However, this study did not find any *Vibrio* species that are always present in YBD lesion but not in healthy corals. Although *Vibrio* have been reported as pathogenic bacteria for various coral diseases, it should be noted that they are a part of common bacterial taxa found in coral microbiomes (*Bourne & Munn, 2005*; *Daniels et al., 2011*; *Gray et al., 2011*; *Nithyanand & Pandian, 2009*). The higher overall abundance of Vibrio in the healthy corals than the infected corals as found in our study could be expected due to the taxonomic refinement of the partial rDNA sequence was analyzed to the genus level only, not specifically to the specific group of pathogenic *Vibrio* species. The conflicting phenomenon on the higher abundance of *Vibrionales* in healthy corals has been previously reported in analysis of symbionts in corals infected by WPD (*Cardenas et al., 2012*; *Kellogg et al., 2013*).

WPD has been reported to affect more than 40 coral species (*Sunagawa et al., 2009*; *Weil, Smith & Gil-Agudelo, 2006*). However, identification of its causative agent is still problematic, suggesting a complex etiological phenomenon involving alterations in the dynamic interaction between environmental factors, and symbiotic microbiomes. According to our study, reduction in the relative abundance of all subphyla of *Proteobacteria* was found in *Isopora* sp. upon infection by WPD with increasing abundance of *Firmicutes* (*Clostridia*) and changes in the phylum *Bacteroidetes* with decreasing *Cytophaga* and increasing *Flavobacteria* along with reduction in *Lentispaheria*. This change in the bacterial community structure led to an increase in *Acrobacter* and WH1-8 bacteria along with the decrease in *Clostridia* and *Fusibacter* at the genus level. Increasing diversity and a shift in bacterial community structure in *Montastraea faveolata* infected by WPS Type II has been shown using high-density 16s rRNA gene microarray and clone library sequencing (*Sunagawa et al., 2009*). Accumulation of various known bacterial families known as coral pathogens including *Alteromonadaceae* and *Vibrionaceae* has been found. However, the primary pathogen *Aurantimonas corallicida* (*Denner et al., 2003*) previously proven by Koch's postulate was not detected in this molecular study (*Sunagawa et al., 2009*). Analysis of differentially abundant OTUs in Caribbean coral species *Orbicella faceolata* and *O. franksi* showed marked differences in bacterial communities in the heathy and diseased coral samples but not between coral species (*Sunagawa, Woodley & Medina, 2010*). The subsequent comparison in Indo-Pacific coral species (*Pavonaduerdeni* and *Poriteslutea*) showed distinct bacterial community patterns associated with ocean basin, coral species and health status. Increasing bacterial richness was found in the diseased samples suggesting the role of opportunistic bacteria during pathogenesis. These studies showed microbial community patterns related to WPD that are consistent over coral species and oceans, irrespective of the putative underlying pathogens (*Sunagawa, Woodley & Medina, 2010*). This includes various taxa involving *Proteobacteria*, *Bacteroidetes*, *Cyanobacteria*, and *Firmicutes*. A higher abundance of known coral pathogens, e.g., *Alteromonadaceae*, *Rhodobacteraceae*, and *Vibrionaceae* have been reported in *Pavona duerdeni* and *Porites lutea* infected by WPD using 16S rRNA gene microarray (*Roder et*

*al., 2014*). A significant increase in *Alphaproteobacteria* and a concomitant decrease in the *Beta-* and *Gammaproteobacteria* were also observed in WPD-affected reef-building corals *Diploriastrigose* and *Siderastrea* sidereal using culture-dependent methods and pyrosequencing of 16s rRNA sequences (*Cardenas et al., 2012*). Significant shifts were also found for the orders *Rhizobiales*, *Caulobacteriales*, *Burkhoderiales*, *Rhodobacteriales*, *Aleteromonadales*, and *Xanthomonadales*, suggesting roles of these bacteria on pathogenesis.

# CONCLUSION

Our findings showed structural alteration of microbiomes associated with important reef-building corals in Indonesian sea revealed by a culture-independent molecular analysis approach. Coral disease pathogenesis led to disturbance of the complex microbial communities and shifts in the overall bacterial community structures as shown by well separated clustering of the heathy and infected coral microbiomes with significant changes in certain key bacterial species. This study provides insights on understanding the functions of associated symbiont bacteria in pathogenesis. Further in-depth analysis on the role of microbiomes on coral health is warrant.

# ACKNOWLEDGEMENTS

The authors would like to thank Dr. Philip J. Shaw for manuscript proofreading and comments.

## Funding

This project was financially supported by the National Center for Genetic Engineering and Biotechnology, National Science and Technology Development Agency, Thailand. The funders had no role in study design, data collection and analysis, decision to publish, or preparation of the manuscript.

## Grant Disclosures

The following grant information was disclosed by the authors:
National Center for Genetic Engineering and Biotechnology.
National Science and Technology Development Agency, Thailand.

## Competing Interests

The authors declare there are no competing interests.

## Author Contributions

- Wuttichai Mhuantong and Pattanop Kanokratana conceived and designed the experiments, performed the experiments, analyzed the data, contributed reagents/materials/analysis tools, prepared figures and/or tables, authored or reviewed drafts of the paper, approved the final draft.

- Handung Nuryadi conceived and designed the experiments, performed the experiments, analyzed the data, contributed reagents/materials/analysis tools, prepared figures and/or tables, approved the final draft.
- Agus Trianto, Agus Sabdono and Lily Eurwilaichitr conceived and designed the experiments, authored or reviewed drafts of the paper, approved the final draft.
- Sithichoke Tangphatsornruang analyzed the data, contributed reagents/materials/analysis tools, authored or reviewed drafts of the paper, approved the final draft.
- Verawat Champreda conceived and designed the experiments, analyzed the data, prepared figures and/or tables, authored or reviewed drafts of the paper, approved the final draft.

## Field Study Permissions

The following information was supplied relating to field study approvals (i.e., approving body and any reference numbers):

Field sampling was approved by the Balai Taman Nasional Karimunjawa (934 /BTNKJ-1.6 /SIMAKSI /2015).

## Data Availability

The 16S rRNA gene sequences are deposited in NCBI Sequence Read Archive (SRA) with the accession number SRP071125.

## Supplemental Information

Supplemental information for this article can be found online at http://dx.doi.org/10.7717/peerj.8137#supplemental-information.

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
