# Peer review of "Comparative analysis of bacterial communities associated with healthy and diseased corals in the Indonesian sea"

_PeerJ, doi:10.7717/peerj.8137_

## Round 0.1 · original submission · Major Revisions

Your manuscript has been evaluated by two reviewers who outlined critical aspects of the article that need to be revised before publication. In particular, numerous grammatical corrections are required. Additionally, improvements for clarity and conciseness are also needed. Please provide a point-by-point rebuttal addressing all of the reviewers comments with your revision.

Reviewer 1 ·

Basic reporting

Overall, the manuscript lacks a clear narrative structure and the text is poorly written, quite prone to grammatical errors and frequently make sentences unclear to the reader. I suggest the authors to reduce wordiness, redundant expressions or phrases. Also, the authors have a tendency to overstate their ideas and/or dismiss references regularly. The connection between phrases and fluency can improve as well.

Experimental design

The amount of information you provide on the methods is not enough to reproduce your experiments. Please be more specific to help others following your protocols. For example, it is not clear for me how you attached your barcodes to your amplicons, did you make a second PCR? or was your PCR done with different primers for each sample? how many different barcodes did you use? Same goes for your sampling and nucleic acid extractions. On the other hand, data analysis was very well explained, thanks for that.

Validity of the findings

Title: no compare metagenomics, this is 16S rDNA amplicon diversity.

L80-85: The authors need to clearly make a distinction of what they are considering metagenomic analysis and analysis based on molecular markers. I suggest sticking to 16S-based diversity instead of calling it metagenomics.

Did you collect healthy tissues from diseased corals? otherwise, how can you take into account variation in individual-specific microbiomes?

Your results are mixing a lot of discussion and for me was hard to understand what you found and contrast it with what others have done. Please revise L165-173 in that regard.

Additional comments

L38: Please rephrase, coral reefs are not niches
L40: This needs references
L42-44: Consider rephrasing to: "The coral reef ecosystem is estimated to provide economic benefit worldwide, accounting for USD 29.8 billion per year from various sectors, e.g. fisheries, tourism, coastal protection, and biodiversity."

L50-52: is wordy and unclear
L59-61: Consider rephrasing to something along these lines: "Many of these diseases are prevalent in reef-builder corals (Order Scleractinia), leading to the deterioration of entire reefs structures"
L62: add "and" to "brown band, red band, and yellow band syndromes are.."
L63-64: This is a very strong statement, either reference it appropriately or tone it down.
L65-68: Try to be concise when giving examples and only emphasize on the most relevant ones. Otherwise, the statement turns lengthy and loses clarity.
L68-70: This statement needs references and examples
L72: remove "by microbiological methods" It is clear that Koch postulates involve microbiological methods. Also, consider replacing "Koch'shypotheis" to "Koch's postulates"
L73-74: Why some of the diseases are capitalized and others are in lowercase. Keep it consistent.
L75-76: either add "the" before "culture-depended" or replace "approach" by "approaches". Also, the phrase is long, is missing commas, and needs a grammar check.
L81-82: replace "take" by "make", replace "any" by "many", and replace "ecological niches" by "ecosystems"
L82: remove "the" and add "approaches" after "culture-independent"
L96: Consider rephrasing to "This work compares coral-associated bacteria in different host species and shows shifts in the bacterial community structure during the diseased stage. Our findings expand the current understanding on the microbiology of most prevalent coral diseases"
L111: "SDS-based DNA extraction" only makes reference to the lysis step you used. Please be more specific and state how you isolated nucleic acids (e.g. phenol-chloroform purification with ethanol precipitation) and importantly specify what modifications you made to previous protocols.
L129: What do you mean that sequences were determined? Do you mean: sequencing was done using the ION PGM platform?
135: rephrase to Data analysis was done in QIIME version 1.9.1
215: replace "as minorities" to "in lower abundances"


Table 2: replace "No. sequence (cleaned)" to "No. of sequences after QC"
Figure 6: it would be good to use colored text or colored boxes as a visual guide to better show your clustering. It's hard to remember all abbreviations, especially if they are not in the figure legend!

Reviewer 2 ·

Basic reporting

Overall, the writing needs to undergo considerable revisions for acceptance. The introduction and discussion should be more cohesive in delivering the message about the relevance of these diseases and their connection to the experimental design of the study (regarding bacterial taxonomy and the connection to studies that have been conducted previously). This is especially relevant because much work has been done on some of these various diseases, and their relation to observations in other geographical locations (previous work) is very interesting. While this connection is not completely absent in the text, the writing can be adjusted in order to better demonstrate the linkages to other previous observations.

I believe this study reveals very important information about the bacterial taxonomy associated with different disease states in various coral species. The overall structuring of the paper could use some revisions as well (see specific comments below). For example, lines 167-172 in the results section are not results from this study but descriptions of the diseases previously described. In addition, much of the grammar and word choices throughout the paper also need to be addressed. I offer some changes below in the general comments section that should be helpful. This study offers important new information in the field of marine disease ecology and I would recommend incorporating these suggestions and other reviewers’ comments for acceptance.

Experimental design

The experimental design is acceptable and research question is very clear. I would recommend using these suggestions to make this more effective in demonstrating relevance and meaning in the scope of connecting these results to the global information on bacterial diseases of the same type. The methods are described with sufficient detail and is replicable.

Validity of the findings

The analyses and results are impactful, especially in the context of the bacterial taxa associated with the four diseases in this study. Most of the statistics are acceptable for the analyses, but many of the specific statistical tests are not reported in the results (e.g. what statistic was run? Tukey, Wilcoxin, T-test?). These types of reporting need to be clearer (see below in general comments).

Additional comments

I suggest the term “bacterial microbiomes” should be changed to “bacterial communities” in the title

Abstract –

First sentence – “global warming” should be changed to “climate change”. “Endangered” is more specific to certain species, and should be changed to “impacted”. Other types of human impacts should be mentioned briefly here, such as overfishing, pollution, etc.

Second sentence – “infected stages” should be changed to “disease states”

The last sentence does not make sense grammatically. “relevant to the pathogenicity of the infected corals”. Maybe change to “pathogenicity of the microbes associated with infected corals”

Main text

Line 38 – Should be changed to “Coral reefs contain important ecological niche space harboring…”

Line 48 – “sponges with closed interactions to associated communities” needs to be elaborated on (i.e. what are these closed interactions?)

Line 51 – “phenomenon” should be plural, see “phenomena”

Line 60 – I would suggest using the word “infect” instead of “attack”, this applies to the rest of the text.

Lines 63-64 – Citations needed

Line 71 – “were” should be “ have been”

Line 76 – Needs to be reworded

Line 78-79 – The final sentence of this paragraph needs to be reworded for clarity

Line 80 – “environmental metagenomes” should be changed to “environmental communities through metagenomic sequencing”

Line 82 – “showed” to “have shown”

Line 86 – Change to “variation in the taxonomic composition”

Line 87 – Change to “existing coral diseases”

Line 93 - Change to “our work”

Line 95-96 – This sentence needs to be restructured

Line 105 – Change to “from different colonies”

Line 160 – Citation needed for the human activities section of this line

Line 167-173 – These are not results from this study but descriptions of diseases that should be included in the discussion

Line 192 – Unclear what metric was used for genetic distance (units?)

Line 195 – What statistical tests are associated with this p-value, this applies to the rest of the text

Line 252 – What statistic was used to determine clustering? Is this just visual?

Figure 3 – Should include the type of disease associated with the corals in the figure, for visual reference. This applies to other figures that meet the same criteria.

---

## Round 0.2 · Minor Revisions

The revised manuscript has been strengthened considerably. Please make the minor corrections outlined by Reviewer 1.

Additionally, please make the following editorial corrections.

In Lines 47-49. Remove "potentially"
Replace "photosynthate, mainly in form of glucose" with "secrete fixed carbon to the coral host".
"For example, dinoflagellate endosymbionts, Symbiodinium potentially utilizes light energy (Brodersen et al., 2014) and secrete photosynthate, mainly in the form of glucose, to the coral host (Burriesci et al., 2012)."
Line 109 - This citation should be Wegley et al., 2007 and also fixed in the References. Linda is the first name.
Line 135 - Remove extra period.
Line 148 - Change Firstly to First.
Lines 169-170. It is customary to list values in the text rather than just describe parameters as higher or lower.
Through-out the results and discussion, please use the whole species name (A. aspera, A.formosa, Cyphastrea sp., Isopora sp.), not just Aa or Af, etc.

Very nice work and I look forward to seeing the final version of the article.

Reviewer 2 ·

Basic reporting

I believe this study reveals very important information about the bacterial taxonomy associated with different disease states in various coral species.

The manuscript still needs minor revisions in my opinion, but the authors addressed most of the major changes I suggested in the first round of review. I would also recommend the other independent reviewer confirm this on their part, or another new reviewer be included in this round.

Experimental design

The experimental design is acceptable and research question is very clear. I would recommend using these suggestions to make this more effective in demonstrating relevance and meaning in the scope of connecting these results to the global information on bacterial diseases of the same type. The methods are described with sufficient detail and is replicable.

Validity of the findings

The analyses and results are impactful, especially in the context of the bacterial taxa associated with the four diseases in this study. The authors addressed most of the critique from the first round of review.

Additional comments

I am satisfied with most of the responses from the rebuttal. I recommend this paper be accepted with additional minor revisions outlined below.

Line 17 – “Coral reef ecosystem” should be plural (e.g. “coral reef ecosystems are impacted by”)
Line 41 – “sea lives” should maybe be changed to “marine organisms”
Line 44 – did you mean “erosion” instead of “abrasion”?
Line 66 – should be “rising” instead of “raising”
Line 106 – consider adding “the Indonesian Sea” instead
Line 313 – “resulted” might be changed to “the result of”
Line 320 – “part of the world” should be changed to “regions of the world”
Line 322 – “finding” should be “findings”
Line 362 – add “,” after Gammaproteobacteria
Line 364 – “co-existed” needs to be defined. I’m not sure what is meant by this statement, or if it is just grammatically incorrect

---

## Round 0.3 · Minor Revisions

Thank you for careful consideration of the reviewer comments. This article will make a fine contribution to the literature. Very nice work!
There are just a couple small comments that arose during the final editorial review which need to be addressed before the article can be accepted.

First, a minor correction to the authors use of "Symbiodinium" - this genus has been split and is no longer valid. The proper name of the group has officially been changed to "Symbiodiniaceae" (LaJeunesse et al. 2018).
Second, given that the Cyphastrea & Isopora in the paper are not identified to species level, it seems important to include specific information that would allow replication of the work. Ideally, the authors would deposit voucher samples to represent the taxa and provide information on where those voucher specimens are deposited. If that is not possible, exact site and details on the unidentified coral colonies (including pictures which would allow for identification, similar to what the authors have done for the diseases) ought to be provided in a supplement for future comparison, replication or clarification.

---

## Round 0.4 · accepted · Accept

Thank you for tending to the editorial comments. I am going to accept the manuscript, but during the final stages of production, please make sure that the new supplementary figure 2 is referenced in the text. I could see from the track changes that Fig S3 and S4 had been modified, but could not find any reference to Fig S2.

Very nice work and congratulations!
Linda